# Influence of 40% Cold Working and Annealing on Precipitation in AISI 316L Austenitic Stainless Steel

**DOI:** 10.3390/ma15186484

**Published:** 2022-09-19

**Authors:** Katarína Bártová, Mária Dománková, Jozef Bárta, Peter Pastier

**Affiliations:** 1Faculty of Materials Science and Technology, Slovak University of Technology, Jána Bottu 25, 917 24 Trnava, Slovakia; 2Welding Research Institute, Račianska 71, 831 02 Bratislava, Slovakia

**Keywords:** austenitic stainless steel, intergranular corrosion, secondary phase precipitation, transmission electron microscopy, EDX analysis, Thermo-Calc

## Abstract

Intergranular corrosion is one of the most important processes affecting the behaviour of austenitic stainless steels. Factors such as steel chemical composition, the degree of prior deformation and the exposure temperature affect the degree of sensitisation. AISI 316L (0% CW) steel was annealed at 650 °C for 5, 10, 30, 100, 300 and 1000 h to analyse the influence of isothermal annealing on the precipitation of secondary phases. AISI 316L steel after 40% cold working and subsequent annealing at 650 °C for 1, 1.5, 2, 5 and 10 h was investigated. Time–temperature sensitisation (TTS) diagrams were created based on corrosion test (ASTM A 262, practice A) results. In the case of AISI 316L (0% CW), M_23_C_6_, chi and sigma phases precipitated at grain boundaries, and the Laves phase was mainly inside of the grains. In the case of AISI 316L (40% CW), sigma, chi, Laves and M_23_C_6_ were identified and precipitated mainly along the grain boundaries as well as on the shear bands within different annealing times. It was confirmed that the increase in the annealing time caused an increase in the amounts of secondary phases. Secondary phases in the equilibrium state were calculated using Thermo-Calc software.

## 1. Introduction

Austenitic stainless steels (ASS) are frequently used as construction materials for various components in the chemical, petrochemical and pharmaceutical industries [1,2,3,4,5]. Moreover, both low- and normal-carbon-grade types of 304 and 316 ASS have been widely used to construct in-core structures, primary pressure boundary components and also auxiliary systems in nuclear power plants. The main reason for this is their combination of excellent mechanical properties (ductility), machinability and corrosion resistance, especially in high-temperature water [6,7,8]. On the other hand, ASS have lower yield strength compared to other stainless steels. Alloy elements in ASS affect the structure, mechanical and corrosion properties. Elements such as Cr, Mo and Si stabilise ferrite, whereas Ni, Mn, N and C stabilise austenite. High contents of Cr and Mo ensure the good corrosion resistance of ASS. Ni stabilises austenite at normal and low temperatures as well as during plastic deformation. It also provides ASS with excellent toughness.

Though resistant to uniform corrosion, ASS are prone to localized corrosion and stress corrosion cracking (SCC) depending on the structural, compositional and morphological characteristics of the grain boundaries, e.g., on the state of intergranular precipitation [9,10,11,12].

When ASS are subjected to long-term exposure to a temperature range of 500 °C to 900 °C or slowly cooled from elevated temperatures, the precipitation of a large number of phases occurs (M_23_C_6_ carbide, sigma, chi and Laves phases) and causes the sensitisation of the steel [1,4,12,13,14,15,16,17,18,19]. The precipitation of chromium-rich phases along the grain boundaries causes the formation of chromium-depleted zones. If the chromium content near the grain boundaries drops under the passivity limit of 12 wt. %, the steel becomes sensitised [9,20]. Sensitisation is the basic reason for intergranular corrosion (IGC) and intergranular stress corrosion cracking (IGSCC) [12].

M_23_C_6_ carbide is usually the first phase to be formed in ASS. Significant amounts of carbides can form after a few minutes in a temperature range of 650 °C to 750 °C, depending on the carbon content. However, in both stabilised and non-stabilised austenitic stainless steels, there is at least partial dissolution of M_23_C_6_ carbide after long ageing times. In non-stabilised steels, the dissolution can also occur if there is significant precipitation of secondary phases such as sigma, chi and Laves phases. M_23_C_6_ carbide precipitates at grain boundaries, followed by incoherent and coherent twin boundaries and finally at dislocations within the grains. Sigma, which is a common phase in AISI 316L steel, precipitates at temperatures ranging from 550 °C to 900 °C at grain boundaries, twin boundaries and triple points, as well as inside austenite grains [17,18,21]. The kinetics of sigma phase precipitation is very slow, so it can take hundreds to thousands of hours for precipitates to form [21]. The chi phase, as a minor phase in AISI 316L steel, was observed to precipitate mainly at grain boundaries, at incoherent and coherent twin boundaries and on dislocations inside of the grains at temperatures close to 750 °C [11]. Chi phase precipitation is faster than the precipitation of the sigma phase due to easier nucleation as well as the ability to dissolve carbon [21]. A long ageing time also led to the precipitation of a minor Laves phase (Fe_2_Mo) in AISI 316L steel with a Mo content of 2–3 wt. %. The Laves phase has regularly shaped particles that precipitate mostly inside of the grains. Sporadically, it can precipitate at the grain boundaries. Cold work (CW), after solution annealing and before ageing, also accelerates the nucleation of M_23_C_6_ carbide, the sigma phase and the chi phase [21].

The chemical composition, cold work degree, grain size and time–temperature exposure have significant effects on the sensitisation of ASS [1,2,4,11]. Time–temperature sensitisation (TTS) diagrams are used to evaluate the influence of these parameters on sensitisation. The TTS diagram has the shape of a C-curve that divides the state of the steel into sensitised or non-sensitised. The nose of this C-curve represents the critical temperature with the minimum time of sensitisation [21].

The main goal of this study was to determine the relations between AISI 316L steel conditions of thermal deformation treatment and the state of precipitation. The primary focus is on the identification of the type, distribution and chemical composition of secondary phases.

## 2. Materials and Methods

The chemical composition of the examined AISI 316L (Böhler, Vienna, Austria) steel is given in Table 1. Solution annealing for 1 h at 1100 °C was applied to the material with subsequent water quenching to eliminate the formation of new precipitates. In this way, treated steel is further referred to as AISI 316L (0% CW). The cold working of AISI 316L (0% CW) was applied at ambient temperature with a deformation of 40% compared to the original sheet thickness. The sample after cold working is further referred to as AISI 316L (40% CW).

AISI 316L (0% CW) steel was annealed using an MLW LM212.11 muffle furnace (VEB Elektro, Berlin, Germany) at 650 °C for 5, 10, 30, 100, 300 and 1000 h to analyse the influence of isothermal annealing on the precipitation of secondary phases. Precipitation is expected after shorter times in austenitic stainless steel after cold work; therefore, AISI 316L (40% CW) steel was annealed at 650 °C for 1, 1.5, 2, 5 and 10 h.

The samples for microstructural analysis were prepared by a standard metallographic procedure (polished up to ~1 μm). Electrolytic etching for 10–30 s in 10% oxalic acid at a 1 A.cm^−2^ current density was applied as a final step. Microstructural analysis was realised using a light microscope (NEOPHOT 32, Carl Zeiss AG, Stuttgart, Germany). To determine the sensitivity to intergranular corrosion, samples were tested according to ASTM A 262, practice A (oxalic acid etch test). The same metallographic procedure, however, with 90 s etching time, was applied during the corrosion test. To evaluate the intergranular corrosion, three criteria were monitored:No corrosion at the grain boundaries (step structure);Grains partially surrounded by deeply etched boundaries (dual structure);Grains completely surrounded by deeply etched boundaries (ditch structure).

Samples meeting criteria 1 or 2 were considered to be non-sensitised. Samples were classified as sensitised when meeting criteria 3 [22].

Carbon extraction replicas were used for transmission electron microscopy (TEM) in order to identify secondary phases. TEM analysis was performed using a JEOL 200 CX microscope with an energy-dispersive X-ray spectrometer (EDX) (JEOL Ltd., Tokyo, Japan). Samples were prepared in several steps to perform TEM analysis. After grinding and polishing, etching in 10 mL of H_2_SO_4_, 10 mL of HNO_3_, 20 mL of HF and 50 mL of H_2_O for 2 to 3 min was realised. Then, a thin layer of carbon was applied to the surface to be further removed from the samples in an 8% solution of HCl in ethanol.

EDX analysis was realised on every type of secondary phase at least 10 times for each sample. In the evaluation of EDX spectra, the standardless method for thin specimens was used.

Thermo-Calc software v.TCW5 (Thermo-Calc Software, version TCW5, Solna, Sweden) [23] using the TCFE6 database was used to calculate the phase equilibria of the system corresponding to the investigated steel in the temperature range of 500–1000 °C. In the calculation procedure, the total Gibbs energy of the system consisting of contributions of individual phases was minimised at constant temperature and pressure. The particular phases were modelled as a sum of the reference levels of Gibbs energy, the entropy term, excess Gibbs energy and the magnetic term (if plausible, the magnetic ordering). The elements C, N, Cr, Ni, Mn, Mo, Si and Fe were considered in calculations, as well as the phases, such as liquid, delta-ferrite (b.c.c.), austenite (f.c.c.), the Laves phase (h.c.p.), M_6_C (f.c.c.), M_23_C_6_ (f.c.c.), M_7_C_3_ (orthorhombic), M_3_C_2_ (orthorhombic), MC (f.c.c.), M_2_N (h.c.p.), the sigma phase (tetragonal) and the chi phase (b.c.c.).

## 3. Results

Polyhedral austenitic grains containing twins were observed in the microstructures of AISI 316L (0% CW) after solution annealing (Figure 1a,c). Cold working caused structural changes, whereas the microstructure analysis revealed acicular grains with occasional shear bands (Figure 1b,d). Grain boundaries were slightly etched, indicating the absence of secondary phases at the grain boundaries in conditions both with and without cold working. Black discrete areas are probably artefacts originating from the etching process, showing an irregular distribution in the microstructure of AISI 316L (40% CW).

### 3.1. Oxalic Acid Etch Test

The TTS diagrams of AISI 316L (0% CW) and AISI 316L (40% CW) steels are illustrated in Figure 2a,b. Each of these diagrams was constructed based on corrosion test results performed on three isotherms. Based on the TTS diagram in Figure 2a, it can be stated that the maximum sensitisation rate (nose of the C-curve) was observed at 750 °C after 10 h in the case of AISI 316L (0% CW) steel. The maximum sensitisation rate of AISI 316L (40% CW) steel was observed at 750°C after 2 h (Figure 2b).

The circles in Figure 2a,b represent the experimental results after the oxalic acid etch test under 650 °C annealing conditions. A step structure (Figure 2a, green circle) was observed in AISI 316L (0% CW) steel after a 5 h annealing time, whereas a dual structure (Figure 2a, blue circles) was observed after 10 h and 30 h. A ditch structure (Figure 2a, red circles) was observed after 100 h, 300 h and 1000 h annealing times, where the sensitisation of steel occurred. The microstructures corresponding to 650 °C/10 h and 650 °C/1000 h conditions are documented in Figure 3a,b.

In AISI 316L (40% CW) steel, a dual structure (Figure 2b, blue circles) was observed at 650 °C after 1 h, 1.5 h, 2 h and 5 h annealing times. The red circle in Figure 2b represents the sensitised condition after 10 h annealing, representing the ditch structure. The microstructures corresponding to 650 °C/1 h and 650 °C/10 h conditions are documented in Figure 3c,d.

### 3.2. TEM Analysis of AISI 316L (0% CW)

AISI 316L (0% CW) steel with annealing conditions of 650 °C/5 h (Figure 4a) had triple points (TP) and grain boundaries (GB) free of precipitates. Longer annealing times led to higher precipitation of secondary phases. Significant amounts of secondary phases at grain boundaries are documented in Figure 4b after a 1000 h annealing time. Four types of secondary phases (Table 2) were identified using electron diffraction analysis.

No secondary phases were observed at 650 °C after a 5 h annealing time. M_23_C_6_ carbide mainly precipitated at grain boundaries after 10 h of annealing, whereas the chi phase was rarely observed. The transition between sensitised and non-sensitised states was determined based on a corrosion test after a 100 h annealing time. In this condition (650 °C/100 h), the Laves phase and, rarely, the sigma phase started to precipitate at grain boundaries. After a 300 h annealing time, M_23_C_6_ carbide, the Laves phase and the sigma phase were identified; however, the chi phase was not identified anymore. Based on this result, it can be concluded that the chi phase is a temporary precipitate. Figure 5 documents examples of secondary phases identified at different annealing times.

M_23_C_6_ carbide was the dominant precipitate and precipitated at grain boundaries in the form of small (short exposure times) as well as massive irregular particles. The presence of chi-phase particles was occasional, present mainly at δ-ferrite grain boundaries. At longer annealing times, the chi phase precipitated at austenite grain boundaries in the form of bigger particles. The Laves phase also precipitated at grain boundaries in the form of small needle-shaped particles. At longer annealing times, the Laves phase was observed to precipitate sporadically at grain boundaries in the form of massive particles. Occasionally, the sigma phase, in the form of small and middle-sized particles, was observed to precipitate close to M_23_C_6_ carbide.

The typical chemical composition of identified secondary phases, together with measurement deviations, are provided in Table 3. At least 100 particles were evaluated for each sample. Differences in the chemical compositions of particular phases based on the annealing time were minimal. These differences varied in the measurement deviations. Based on these results, the average chemical composition of each secondary phase for the 650 °C isotherm was determined.

### 3.3. TEM Analysis of AISI 316L (40% CW)

To identify the types of secondary phases after annealing, TEM analysis was carried out. Figure 6a shows that typical grain boundaries (GB) formed at an early stage of annealing (650 °C/1 h), and only a few secondary phases were observed on shear bands (SB). In general, the amounts of secondary phases in the microstructure were found to increase with increasing annealing time [11,13,21]. When annealing at 650 °C/10 h (Figure 6b), secondary phases were observed to precipitate on the shear bands, on the grain boundaries, in triple points (TP) and inside the grains.

Four types of secondary phases were identified, and the results are summarised in Table 4. The sigma phase at shear bands was detected after annealing, and it was identified in all analysed conditions. The precipitation of the chi phase was observed after 2 h, and M_23_C_6_ carbide was detected after 5 h of annealing. These transitions did not influence the steel’s resistance to intergranular corrosion (Figure 2b). The important transition between non-sensitised and sensitised states was determined after 5–10 h of annealing at 650 °C according to the corrosion test ASTM 262, practice A (Figure 2b). In this transition, the precipitation of the Laves phase was observed.

The sigma phase was observed mainly on the shear bands (Figure 7a) and at the grain boundaries. The chi phase (Figure 7b), the Laves phase (Figure 7c) and M_23_C_6_ carbide (Figure 7d) were found to precipitate along the grain boundaries. Moreover, Laves-phase particles were also identified inside the grains.

The annealing time did not significantly influence the average values of the chemical composition of the identified phases. Therefore, the average chemical compositions of particular phases, regardless of the annealing conditions, were calculated (Table 5).

### 3.4. Thermodynamic Prediction

The equilibrium phase diagram (Figure 8) for the investigated steel was calculated using Thermo-Calc software for the temperature range of 500 °C to 1000 °C.

The mole fractions of the phases present in AISI 316L steel (Figure 9) as a function of the annealing temperature were also calculated using Thermo-Calc software. Besides austenite forming the matrix, the sigma phase, M_23_C_6_ carbide and M_2_N nitride were calculated in the equilibrium state at 650 °C.

The influence of the annealing temperature on the chemical composition of predicted equilibrium secondary phases was calculated using Thermo-Calc software. The change in the chemical composition of M_23_C_6_ carbide (Figure 10a) was insignificant in the selected temperature range; however, at higher temperatures, the amount of Fe rose. More significant differences were observed in the sigma phase (Figure 10b), where the Cr content decreased with increasing temperature, and, conversely, the Mo content increased.

The chemical compositions of M_23_C_6_ carbide (66.7% Cr, 8.3% Fe, 19.1% Mo, 0.6% Ni, 0.2% Mn, 0.2% Si, 0.5% N and 4.6% C) and the sigma phase (36.1% Cr, 51.3% Fe, 9.4% Mo, 3.4% Ni and 0.5% Mn) were calculated using Thermo-Calc software at a 650 °C annealing temperature.

## 4. Discussion

The susceptibility of AISI 316L to intergranular corrosion was evaluated by the oxalic acid etch test. TTS diagrams were created based on the results of this test (Figure 2a,b) for AISI 316L (0% CW) and AISI 316L (40% CW), respectively. Table 6 summarises the parameters of C-curves’ critical areas, meaning the critical temperature (T_crit_), where sensitisation occurs in the shortest annealing time (t_min_). The influence of 40% cold working on the sensitisation of AISI 316L was determined based on these parameters.

Parvathavarthini et al. [1] studied the sensitisation of AISI 316LN steel (0.043% C, 17.18% Cr, 10.23% Ni, 1.85% Mo and 0.075% N). They set the minimum time necessary to sensitise the steel to t_min_ ~ 4.4 h at a critical temperature T_krit_ of ~ 650 °C. In comparison to our results, the differences may be caused by the higher C and N contents in AISI 316LN. In a different paper, Parvathavarthini et al. [24] set t_min_ for AISI 316L steel (0.02% C, 18.5% Cr, 11.5% Ni, 2.3% Mo and 0.07% N) to 40 h at a critical temperature of 650 °C. The longer sensitisation time for their AISI 316L (40 h) in comparison to our AISI 316L (10 h) may be caused by the higher N content. Parvathavarthini et al. [1] investigated the effect of the transformation degree on the sensitisation of AISI 316LN steel (0.043% C, 17.18% Cr, 10.23% Ni, 1.85% Mo and 0.075% N). The critical temperature and minimum time required to sensitise this steel after 25% deformation were 680 °C and 3.6 h. The differences compared to our results could be caused by the lower degree of deformation and increased C and N contents in AISI 316LN steel.

After annealing at 650 °C for 5 h, clear grain boundaries without the presence of precipitated secondary phases were observed in AISI 316L (0% CW). With the extension of the annealing time to 10 h, the precipitation of M_23_C_6_ carbide was observed at triple points and along the austenitic grain boundaries. Rarely, a chi phase precipitating mainly near the δ-ferrite and austenite-phase interface was observed. All four types of secondary phases, as well as sensitisation, were observed in the sample after a 100 h annealing time. After a 300 h annealing time, M_23_C_6_ carbide, the Laves phase and the sigma phase were identified; however, the chi phase was not identified anymore. Based on these results, it can be concluded that the chi phase is a temporary precipitate. Matula et al. [16] identified M_23_C_6_ carbide and the Laves phase after a 500 h annealing time at 650 °C in AISI 316L steel (0.016% C, 16.7% Cr, 10.3% Ni, 2.1% Mo and 0.067% N). The authors rarely observed M_6_C carbide or the chi phase along the grain boundaries after a long annealing time. Sahlaoui et al. [25] identified M_23_C_6_ carbide as the first phase after 40 h exposure in AISI 316L steel (0.022% C, 17.3% Cr, 13.4% Ni, 2.13% Mo and 0.035% N), followed by the Laves phase and the sigma phase for longer exposures. Parvathavarthini et al. [1] confirmed the presence of secondary phases in AISI 316LN steel (0.043% C, 17.18% Cr, 10.23% Ni, 1.85% Mo and 0.075% N) with 0% to 25% CW. M_23_C_6_ carbide, the chi phase and the Laves phase were identified for 0% CW steel after a 500 h annealing time at 650 °C. These results are partially consistent with our results.

Regarding AISI 316L (40% CW) steel, four types of secondary phases were identified at an annealing temperature of 650 °C by TEM analysis. It can be concluded that the amounts of precipitated secondary phases in the microstructure at this temperature increased with increasing exposure time, which is consistent with the conclusions of other authors [11,13,21]. The sigma phase precipitated first on shear bands after a 1 h annealing time at 650 °C. Subsequently, the chi phase and M_23_C_6_ carbide started to precipitate after 2 h and 5 h, respectively. The Laves phase was identified after 10 h, at which time the transition between the sensitised and non-sensitised states was determined based on a corrosion test. Comparing these results with other authors’ data is difficult because the authors examined steels either undeformed or less deformed (up to 25% deformation). Besides M_23_C_6_ carbide, the chi phase and the Laves phase, which were present in the 0% CW state of AISI 316LN steel (0.043% C, 17.18% Cr, 10.23% Ni, 1.85% Mo and 0.075% N), Parvathavarthini et al. [1] also identified carbonitride at the 25% CW state after a 500 h annealing time at 650 °C. However, the presence of phases was not examined at times shorter than 500 h.

The thermodynamic predictions of phase equilibria for the system corresponding to AISI 316L steel showed partial agreement with the experimental results of AISI 316L (0% CW). According to the prediction, M_23_C_6_ carbide, M_2_N nitride and the sigma phase are the secondary phases co-existing with austenite in equilibrium at 650 °C (Figure 8). Experimental confirmation of the presence of M_23_C_6_ carbide and the sigma phase after a 1000 h annealing time is in accordance with the results from the Thermo-Calc software. The chi phase was not identified during longer annealing times, so we can consider it a non-equilibrium intermetallic phase (temporary precipitate), which is in agreement with the software prediction. According to the results, it can be concluded that an annealing time of 1000 h is not sufficient to reach the equilibrium state of the steel. Based on software calculations, we assume that at longer annealing times, the Laves phase will not be identified anymore, and the M_2_N nitride will start to precipitate. Therefore, the amounts of M_23_C_6_, M_2_N and sigma phases in equilibrium should increase at the expense of non-equilibrium intermetallic phases.

The comparison of the secondary phases’ chemical compositions measured in the AISI 316L (0% CW) sample with the values calculated using the Thermo-Calc software showed significant agreement. In the case of the sigma phase, the values of Fe, Mo and Ni are almost identical. The differences are within the range of the EDX analysis measurement error. A small difference was observed in the Cr value of the sigma phase. The Cr values for M_23_C_6_ carbide were almost identical; however, certain differences were observed regarding Fe and Mo values. We were not able to accurately measure the N and C values by EDX analysis because these are light elements; moreover, a TEM carbon replica was used as a sample for the EDX analysis, and therefore, carbon content measurement is not applicable. 

Kherrouba et al. [26] determined the sigma phase chemical composition (50.98% Fe, 30.50% Cr, 9.71% Ni and 13.25% Mo) in AISI 316Ti steel (800°C/8 h annealing conditions) using EDX analysis. They compared this composition with the simulation results of MatCalc (46.43% Fe, 39.39% Cr, 5.09% Ni and 8.83% Mo) and found that the sigma phase was rich in Cr and Mo. Their measured values of the sigma phase’s chemical composition are consistent with our measurement. Regarding the predicted values of the sigma phase’s chemical composition calculated by MatCalc software, similarities in Mo and Ni values were observed; however, the differences in Fe and Cr values were more significant. Ben Rhouma et al. [27] used EDX analysis to measure the chemical composition of the sigma phase (50.0% Fe, 35.8% Cr, 2,7% Ni, 8.7% Mo, 2.0% Mn and 0.8% Si), chi phase (50.0% Fe, 25.0% Cr, 3.0% Ni, 18.1% Mo, 2.5% Mn and 1.4% Si) and Laves phase (27.6% Fe, 33.0% Cr, 4.5% Ni, 30.0% Mo, 1.4% Mn and 3.5% Si) for AISI 316L steel, which was exposed to a temperature of 650 °C for 10,000 h. The measured values of the chemical composition of the sigma phase are similar to our values. On the other hand, significant differences in Fe, Cr and Mo contents could be observed in the chi phase and Laves phase. The chemical composition of secondary phases in AISI 316L (650 °C/30,000 h) steel was published by Sahlaoui et al. [18]. M_23_C_6_ carbide precipitated at the grain boundaries (14.1% Fe, 73.0% Cr, 2.9% Ni, 8.6% Mo, 1.2% Mn and 0.2% Si) and inside the grains (15.1% Fe, 61.5% Cr, 3.9% Ni, 16.6% Mo, 2.5% Mn and 0.4% Si), while differences in Cr and Mo contents were noticed. The sigma phase (47.7% Fe, 38.5% Cr, 3.4% Ni, 8.8% Mo, 1.1% Mn and 0.5% Si) precipitated at the grain boundary, close to M_23_C_6_ carbide and the Laves phase (41.7% Fe, 17.5% Cr, 3.8% Ni, 34.5% Mo, 0.7% Mn and 1.8% Si). In the case of carbide, their results are in good agreement with ours, but significant differences are noticeable in the sigma phase and Laves phase. This could be due to the fact that the steel was annealed for 30,000 h.

## 5. Conclusions

The present work deals with the secondary phase precipitation’s influence on the intergranular corrosion resistance of AISI 316L steel before and after 40% CW with subsequent annealing at 650 °C. Based on the experimental results, several conclusions can be stated:Longer annealing times led to the intensive precipitation of secondary phases at the grain boundaries as well as inside the austenitic grains.A significant decrease in resistance to intergranular corrosion after the application of 40% CW to AISI 316L steel was observed.Four types of secondary phases (M_23_C_6_ carbide, sigma phase, chi phase and Laves phase) present in both AISI 316L (0% CW) and AISI 316L (40% CW) steels at a 650 °C temperature were experimentally confirmed by electron diffraction.The types of secondary phases could not be recognised based on their morphologyCold work of 40% had a significant influence on secondary phases’ precipitation kinetics. The most significant influence was observed on the sigma phase, where the beginning of precipitation was more than 50 times accelerated. A lower influence was observed on the Laves phase and chi phase, where the acceleration was only 5 times higher. The lowest influence of cold work on the precipitation kinetics was observed in the case of M_23_C_6_ carbide.The influence of the annealing time and 40% CW on the chemical composition of secondary phases was not observed.The phase composition predicted by Thermo-Calc software for a 650 °C temperature in the equilibrium state contained austenite, M_23_C_6_ carbide, M_2_N nitride and sigma phase, which is partially in line with experimental results.

## Figures and Tables

**Figure 1 materials-15-06484-f001:**
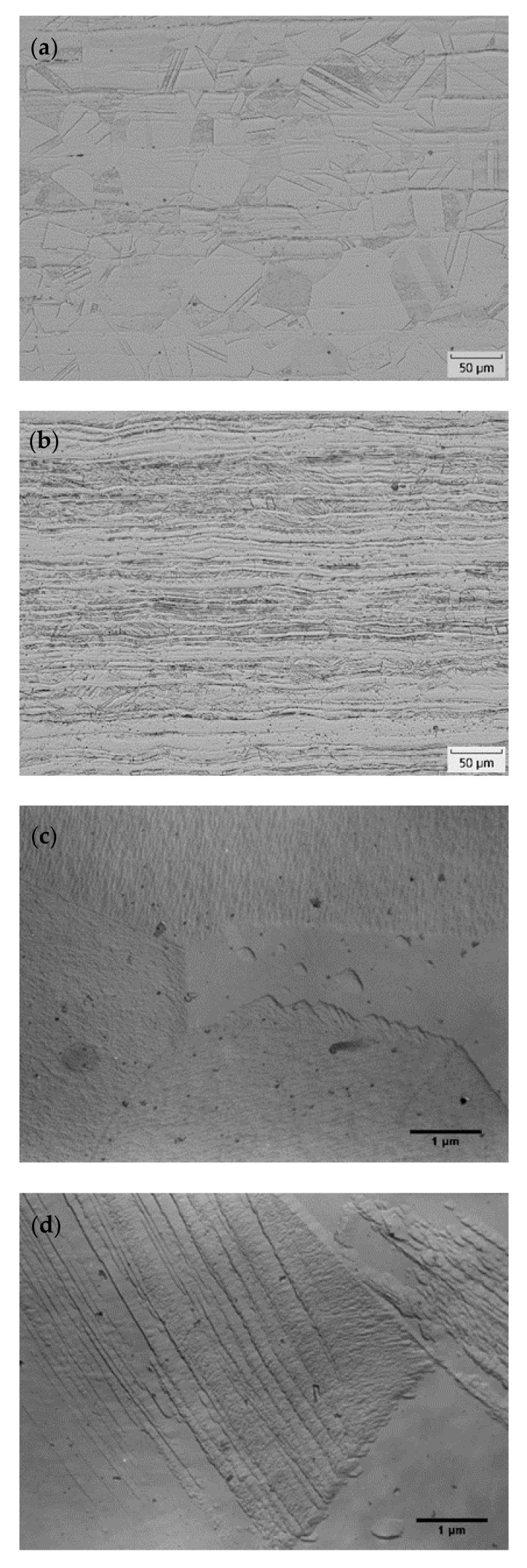
Microstructures of AISI 316L: (**a**) AISI 316L (0% CW)—light microscopy; (**b**) AISI 316L (40% CW)—light microscopy; (**c**) AISI 316L (0% CW)—TEM of a carbon replica; (**d**) AISI 316L (40% CW)—TEM of a carbon replica.

**Figure 2 materials-15-06484-f002:**
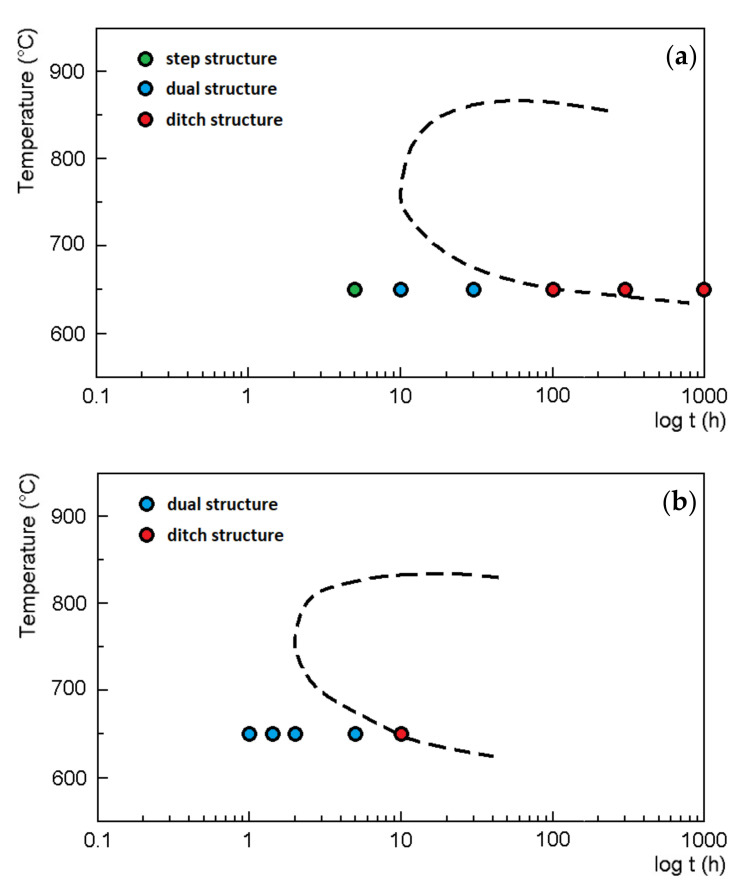
TTS diagrams of AISI 316L: (**a**) AISI 316L (0% CW); (**b**) AISI 316L (40% CW).

**Figure 3 materials-15-06484-f003:**
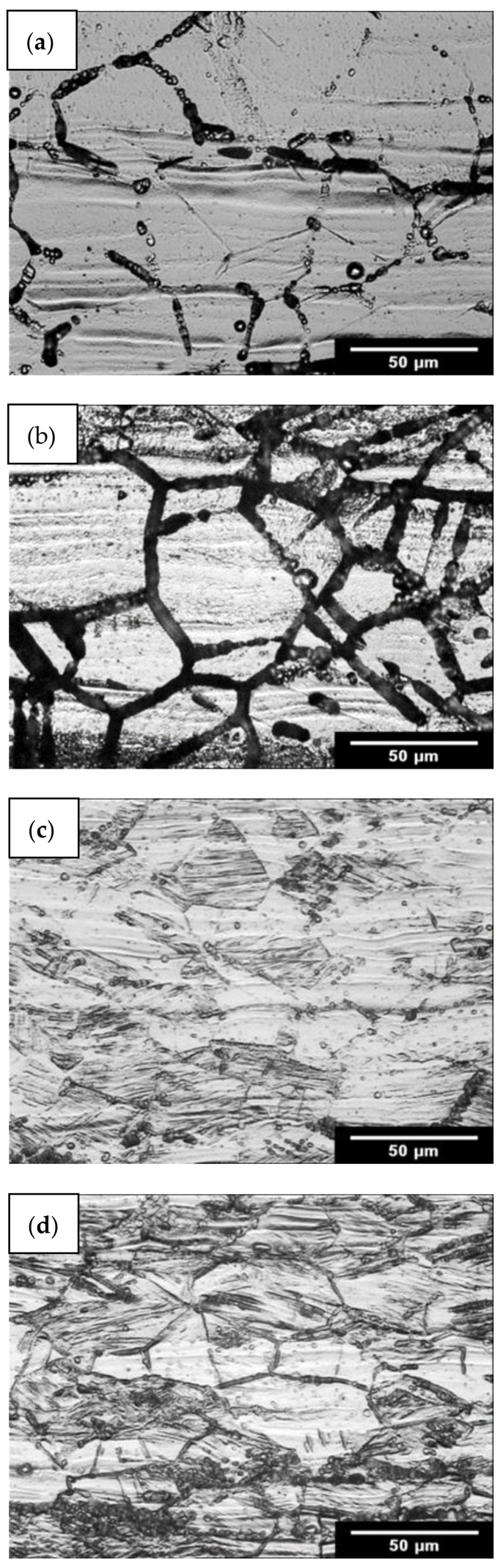
Results of oxalic acid etch test for AISI 316L: (**a**) AISI 316L (0% CW) dual structure observed in 650 °C/10 h conditions; (**b**) AISI 316L (0% CW) ditch structure observed in 650 °C/1000 h conditions; (**c**) AISI 316L (40% CW) dual structure observed in 650 °C/1 h conditions; (**d**) AISI 316L (40% CW) ditch structure observed in 650 °C/10 h conditions.

**Figure 4 materials-15-06484-f004:**
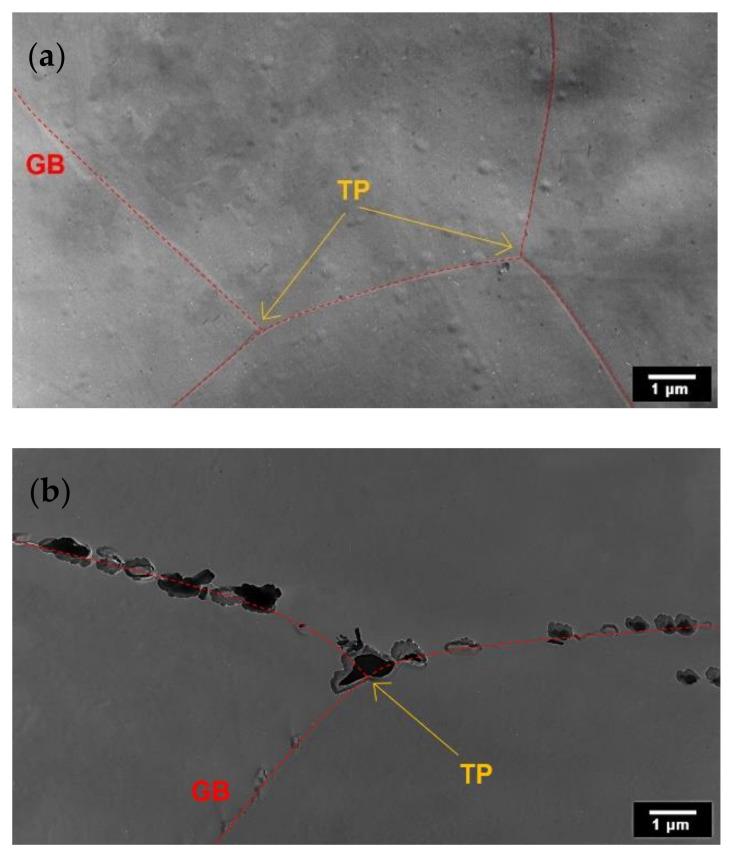
TEM analysis results of carbon extraction replicas of AISI 316L (0% CW): (**a**) 650 °C/5 h of annealing; (**b**) 650 °C/1000 h of annealing.

**Figure 5 materials-15-06484-f005:**
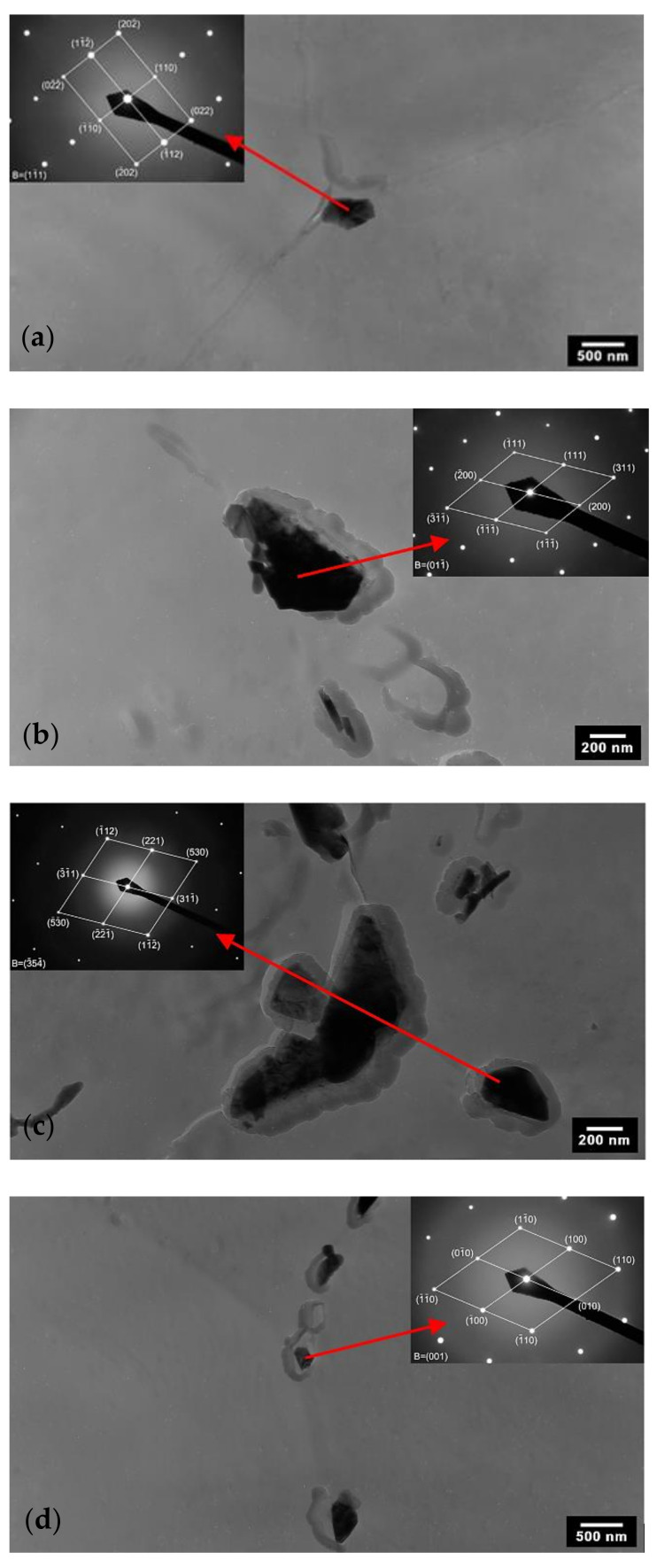
Secondary phases identified in AISI 316L (0% CW) with diffraction patterns corresponding to the marked particles (TEM of carbon extraction replicas): (**a**) chi phase in 650 °C/10 h conditions; (**b**) M_23_C_6_ carbide in 650 °C/1000 h conditions; (**c**) sigma phase in 650 °C/1000 h conditions; (**d**) Laves phase in 650 °C/1000 h conditions.

**Figure 6 materials-15-06484-f006:**
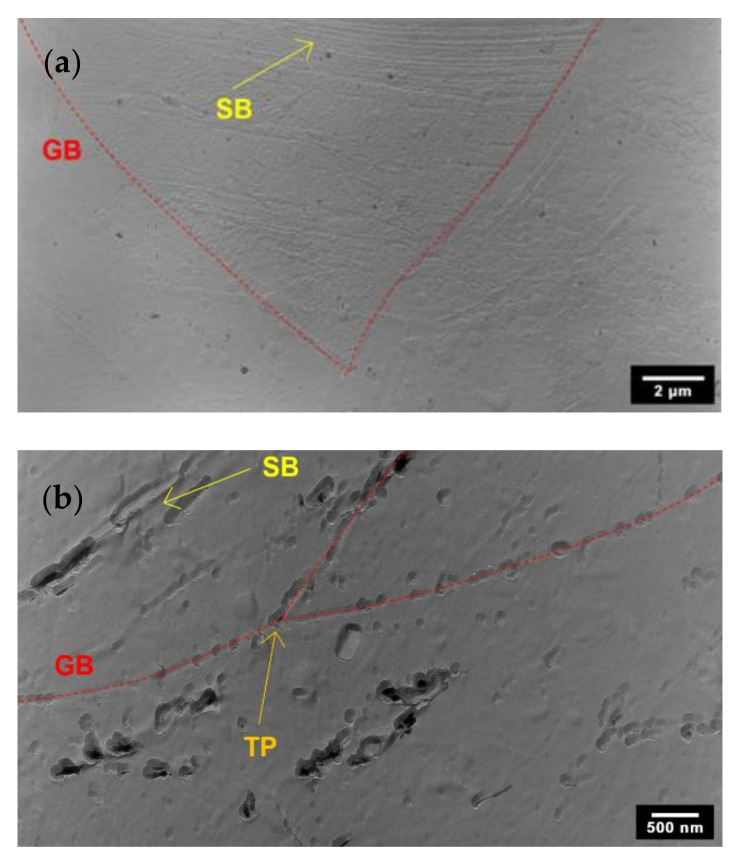
TEM analysis results of carbon extraction replicas of AISI 316L (40% CW): (**a**) 650 °C/1 h of annealing; (**b**) 650 °C/10 h.

**Figure 7 materials-15-06484-f007:**
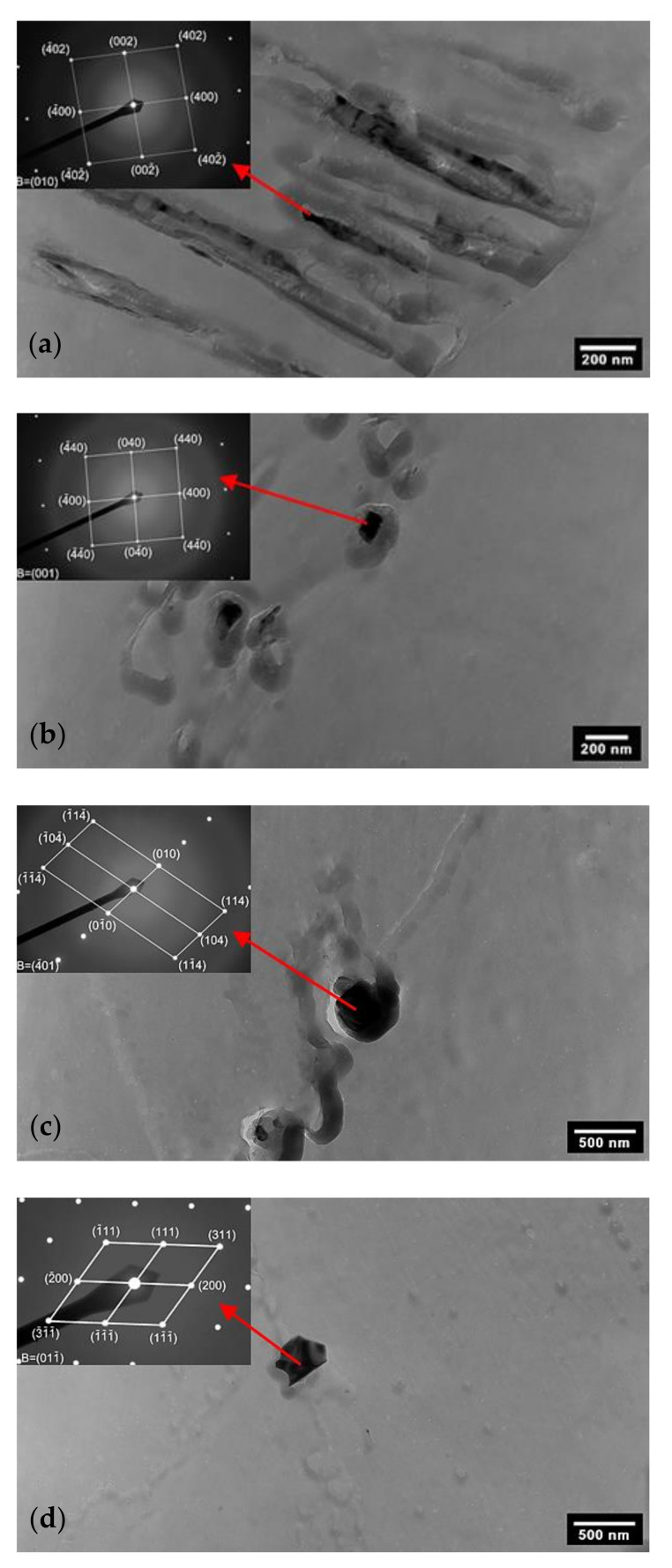
Secondary phases identified in AISI 316L (40% CW) with diffraction patterns corresponding to the marked particles (TEM of carbon extraction replicas): (**a**) sigma phase in 650 °C/5 h conditions; (**b**) chi phase in 650 °C/10 h conditions; (**c**) Laves phase in 650 °C/10 h conditions; (**d**) M_23_C_6_ in 650 °C/10 h conditions.

**Figure 8 materials-15-06484-f008:**
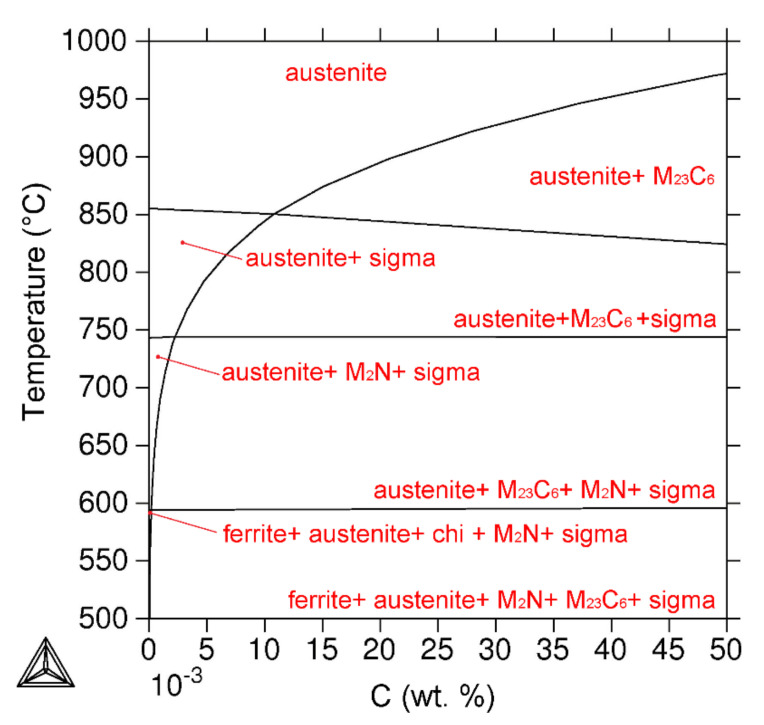
Equilibrium phase diagram for AISI 316L steel.

**Figure 9 materials-15-06484-f009:**
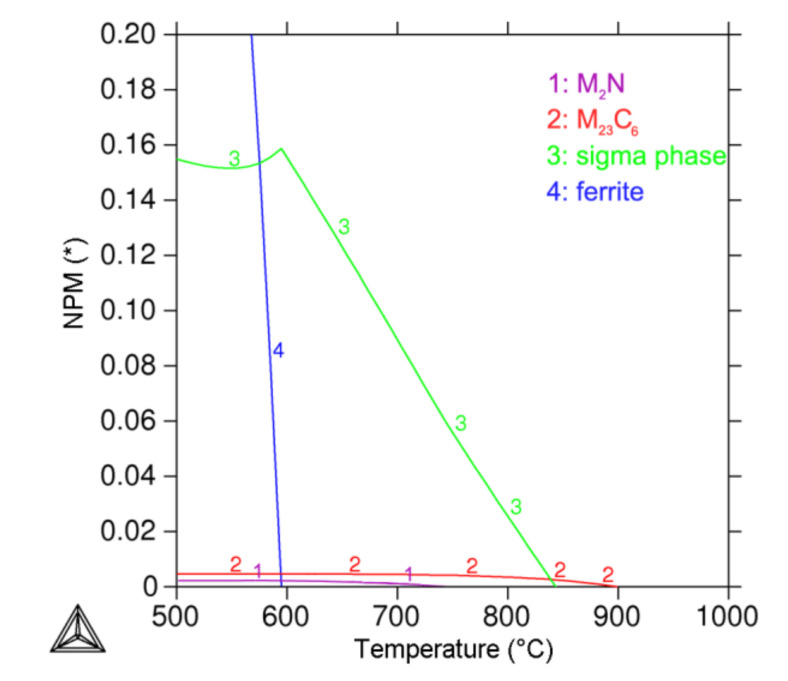
Mole fraction of phases in AISI 316L steel as a function of temperature.

**Figure 10 materials-15-06484-f010:**
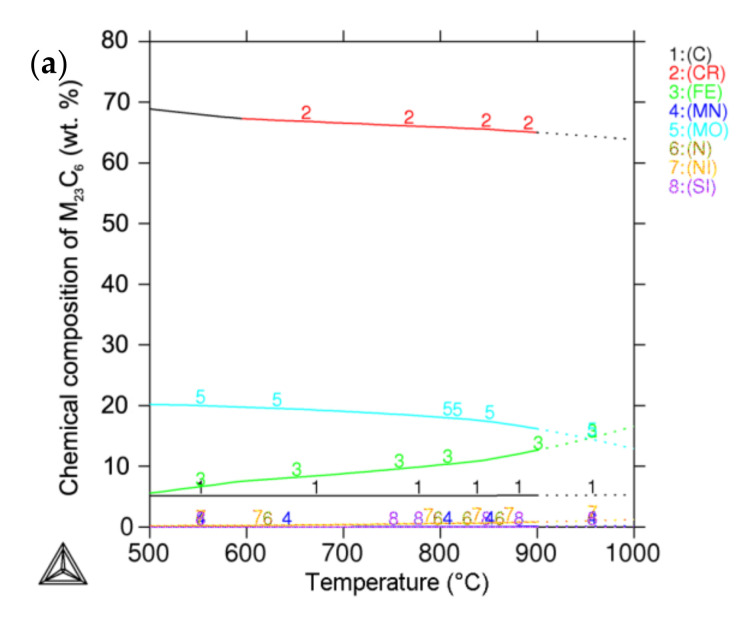
Influence of annealing temperature on chemical composition of predicted secondary phases in AISI 316L steel: (**a**) M_23_C_6_ carbide; (**b**) sigma phase.

**Table 1 materials-15-06484-t001:** Chemical composition of AISI 316L in wt. %.

C	N	Si	Mn	P	S	Cr	Ni	Mo	Ti	Fe
0.021	0.019	0.62	1.10	0.0027	0.004	17.47	12.20	2.10	-	bal.

**Table 2 materials-15-06484-t002:** Experimentally identified secondary phases in the investigated AISI 316L steel (0% CW) with subsequent annealing at 650 °C.

Sample Nr.	Annealing Time (h)	Sigma Phase	Chi Phase	M_23_C_6_ Carbide	Laves Phase
L1	5	-	-	-	-
L2	10	-	×	×	-
L3	30	-	×	×	-
L4	100	×	×	×	⊗
L5	300	×	-	×	×
L6	1000	×	-	×	×

- Secondary phases were not identified; × secondary phases were identified by electron diffraction and EDX analysis; ⊗ secondary phase was measured by EDX analysis but was not confirmed by electron diffraction

**Table 3 materials-15-06484-t003:** Chemical compositions of identified secondary phases in AISI 316L (0% CW) with conditions provided in Table 2.

Secondary Phases	Sample Nr.	Cr	Fe	Mo	Ni	Si
M_23_C_6_ carbide	L2	67.0 ± 1.5	17.7 ± 1.6	12.4 ± 1.8	2.2 ± 0.5	0.6 ± 0.3
	L3	67.2 ± 1.3	17.1 ± 1.3	13.2 ± 1.1	2.0 ± 0.7	0.5 ± 0.2
	L4	68.8 ± 2.0	15.0 ± 1.3	12.8 ± 2.1	2.8 ± 0.4	0.6 ± 0.2
	L5	68.2 ± 1.5	15.4 ± 2.0	13.8 ± 1.7	2.1 ± 0.6	0.5 ± 0.3
	L6	67.3 ± 1.9	15.9 ± 2.5	14.5 ± 1.7	2.0 ± 0.5	0.4 ± 0.2
Average Chemical Composition of M_23_C_6_		67.7	16.2	13.3	2.2	0.5
Chi Phase	L2	21.8 ± 1.4	38.1 ± 1.3	34.7 ± 0.8	2.9 ± 0.5	2.5 ± 0.4
	L3	19.8 ± 1.5	39.9 ± 1.5	35.1 ± 1.1	2.5 ± 0.7	2.7 ± 0.4
	L4	17.5 ± 1.8	39.1 ± 1.3	37.9 ± 1.9	2.3 ± 0.4	3.2 ± 0.3
Average Chemical Composition of Chi Phase		19.7	39.0	35.9	2.6	2.8
Laves Phase	L4	11.6 ± 1.7	30.3 ± 2.1	50.5 ± 1.8	3.0 ± 1.1	4.6 ± 0.6
	L5	10.1 ± 1.2	29.3 ± 1.3	52.1 ± 1.9	3.9 ± 0.8	4.6 ± 0.7
	L6	11.7 ± 1.3	28.8 ± 2.3	50.7 ± 2.5	3.6 ± 0.6	5.2 ± 1.4
Average Chemical Composition of Laves Phase		11.1	29.5	51.1	3.5	4.8
Sigma Phase	L4	32.5 ± 0.9	53.1 ± 0.2	9.6 ± 1.6	2.7 ± 0.3	2.1 ± 0.6
	L5	30.4 ± 0.7	52.0 ± 0.5	12.6 ± 0.9	3.0 ± 0.4	2.0 ± 0.5
	L6	32.3 ± 0.5	51.8 ± 0.9	10.5 ± 0.6	3.2 ± 0.4	2.4 ± 0.6
Average Chemical Composition of Sigma Phase		31.7	52.3	10.9	3.0	2.2

**Table 4 materials-15-06484-t004:** Experimentally identified secondary phases in the investigated AISI 316L steel (40% CW) with subsequent annealing at 650 °C.

Sample Nr.	Annealing Time (h)	Sigma Phase	Chi Phase	M_23_C_6_ Carbide	Laves Phase
DL1	1	×	-	-	-
DL2	1.5	×	-	-	-
DL3	2	×	×	-	-
DL4	5	×	×	×	-
DL5	10	×	×	×	×

- Secondary phases were not identified; × secondary phases were identified by electron diffraction and EDX analysis.

**Table 5 materials-15-06484-t005:** Chemical compositions of identified secondary phases in AISI 316L (40% CW) with conditions provided in Table 4.

Secondary Phases	Sample Nr.	Cr	Fe	Mo	Ni	Si
Sigma Phase	DL51	23.9 ± 2.0	60.2 ± 3.2	5.7 ± 0.7	8.1 ± 0.5	2.1 ± 0.7
	DL52	26.3 ± 1.6	60.3 ± 2.9	8.7 ± 2.1	2.9 ± 1.2	1.8 ± 0.4
	DL53	26.3 ± 1.7	58.5 ± 3.1	6.8 ± 1.7	2.7 ± 1.0	5.7 ± 2.1
	DL54	26.4 ± 1.5	56.2 ± 2.0	9.8 ± 1.9	4.9 ± 1.6	2.8 ± 0.8
	DL55	28.7 ± 1.9	56.3 ± 2.4	10.0 ± 2.4	3.6 ± 1.0	1.3 ± 0.6
Average Chemical Composition of Sigma Phase		26.3 ± 1.5	58.3 ± 1.8	8.2 ± 1.7	4.4 ± 2.0	2.7 ± 1.6
Chi Phase	DL53	23.2 ± 0.8	34.9 ± 1.2	32.9 ± 1.7	3.9 ± 0.9	5.0 ± 1.4
	DL54	24.3 ± 1.0	38.9 ± 2.2	29.0 ± 1.2	4.4 ± 2.4	3.5 ± 2.3
	DL55	24.0 ± 1.8	40.5 ± 2.3	28.5 ± 2.3	4.0 ± 1.7	3.1 ± 0.9
Average Chemical Composition of Chi Phase		23.8 ± 0.5	38.1 ± 2.4	30.1 ± 2.0	4.1 ± 0.2	3.9 ± 0.8
M_23_C_6_ carbide	DL54	62.0 ± 3.0	18.3 ± 1.8	12.4 ± 2.7	4.9 ± 1.4	2.4 ± 0.6
	DL55	64.5 ± 2.4	16.2 ± 1.9	15.6 ± 1.9	2.8 ± 2.0	0.9 ± 0.3
Average Chemical Composition of M_23_C_6_		63.3 ± 1.3	17.3 ± 1.1	14.0 ± 1.6	3.9 ± 1.1	1.7 ± 0.8
Laves Phase	DL55	11.3 ± 1.1	17.0 ± 0.6	54.5 ± 2.6	11.1 ± 1.9	6.2 ± 1.2

**Table 6 materials-15-06484-t006:** Parameters of C-curves’ critical areas T_crit_ and t_min_ for particular TTS diagrams.

ASS	t_min_ (h)	T_crit_ (°C)
AISI 316L (0% CW)	10	750
AISI 316L (40% CW)	2	750

## Data Availability

Not applicable.

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
