# Peer review of "Influence of 40% Cold Working and Annealing on Precipitation in AISI 316L Austenitic Stainless Steel"

_materials, 2022, doi:10.3390/ma15186484_

Round 1

Reviewer 1 Report

The paper ”Influence of 40 % cold working and annealing on precipitation in AISI 316L austenitic stainless steel” has an important set of analysis in the field of AISI 316L. But, in order to be publish, the article needs some minor corrections:

- after line 69, in introduction the authors should add the influence of chemical elements of 316L for mechanical and corrosion properties.

-line 90: add the heat treatment facility

-line 174: please XRD analysis, in order to confirm the mentioned phases

- the same with the other tables

- synthetise the conclusions

- improve the reference list: 20 references are not enought .

In rest is ok.

Author Response

Thank you for your remarks. Construction criticism is always beneficial to improve the quality of the paper. We attached our statements right beneath your comments, marked as bold text. 

- after line 69, in introduction the authors should add the influence of chemical elements of 316L for mechanical and corrosion properties.
The influence of chemical elements was added to introduction 

-line 90: add the heat treatment facility

Muffle oven has been added to the text.

-line 174: please XRD analysis, in order to confirm the mentioned phases

Secondary phases were identified by electron diffraction. Chemical composition of secondary phases was measured by energy dispersive X-ray spectroscopy (EDX). It means that semi-quantitative chemical microanalysis was used. Regarding Table 2, in case of Laves phase (sample L4) was not confirmed by electron diffraction, however we were able to measure chemical composition corresponding to Laves phase. 

- the same with the other tables

- synthetise the conclusions

Conclusions were reorganised in order to make them short and clear.

- improve the reference list: 20 references are not enought .

Another 7 citations were added.

Reviewer 2 Report

Austenitic stainless steels (ASS) are frequently used as construction materials of various components in chemical, petrochemical, pharmaceutical. Moreover, both low and normal carbon grade types of 304 and 316 ASS, have been widely used to construct in-core structure, primary pressure boundary components and also auxiliary systems in 33 nuclear power plants because of their combination of good mechanical and corrosion resistance performance, especially in high temperature water. The paper deals with “Influence of 40 % cold working and annealing on precipitation 2 in AISI 316L austenitic stainless steel”, some results have been obtained in this paper. However, the current version still needs to be greatly improved as follows,

(1)    Why only 40% deformation was chosen in this paper?

(2)    The TEM quality is poor, it should be improved to clearly show the precipitate distribution.

(3)    The initial microstructure should be also characterized by TEM? Additionally, the precipitates in the grains should be also characterized, because the cold deformation also gives a effect on them.

Author Response

Thank you for your remarks. Construction criticism is always beneficial to improve the quality of the paper. We attached our statements right beneath your comments, marked as bold text.

(1) Why only 40% deformation was chosen in this paper?

In previous research different deformations degrees were evaluated (10%, 20%, 40%). Biggest influence on corrosion resistance of AISI 316L had 40% cold work. This paper is therefore focused on precipitation kinetics of secondary phases at 40% cold work. This research was also realised not only at 650°C but also at 750°C and 900°C. Providing all results would extend the paper too much.

(2) The TEM quality is poor, it should be improved to clearly show the precipitate distribution.

We did our best to improve the quality of all figures, however in cases of carbon replicas, the figures are not as good as in case of thin foils. We could not use thin foils otherwise we could not realise EDX analysis.

(3) The initial microstructure should be also characterized by TEM? Additionally, the precipitates in the grains should be also characterized, because the cold deformation also gives a effect on them.

AISI 316L microstructure after solution annealing and 40% cold work was also observed by TEM. Documentation was added to the paper. Precipitates inside of the grains were also characterised however, we made a mistake, providing same figures at figure 4 and 9. Figure 9 was replaced and you can see the microstructure of sample after deformation which contain secondary phases precipitating on shear bands.

Reviewer 3 Report

The peer-reviewed paper examines the effect of heat treatment (annealing) and cold working on microstructure formation and secondary phase precipitation. The work is devoted to topical issues and may be of interest to other researchers. However, I have remarks to the authors of the paper. I hope my comments will help them to improve the paper and publish it in this highly rated scientific journal.

1. There are very few modern studies in the reference. Many of the works cited by the authors were completed more than 20 years ago. It is necessary to analyze modern publications, and there are quite a lot of them. Here are just a few of them:

https://doi.org/10.3390/met12081319

https://doi.org/10.3390/ma15082784

https://doi.org/10.3390/ma15072468

https://doi.org/10.3390/ma15010084

https://doi.org/10.3390/ma15124057

2. Photo in Fig.1 low resolution. It is necessary to improve the quality of the photo and indicate on it the shear bands, grain boundaries.

3. I recommend combining the diagrams in Fig. 2 (a) and 3 (a) into one Fig. 2 (a) and (b) and bringing them side by side so that the reader can compare them and better see the differences. Combine photos of microstructures in Fig.3.

4. In Fig. 4, you need to add indicators of triple points, boundaries of grains of secondary segregations.

5. The photos in Fig. 4 and 6 are the same, but the captions of the figures are different. This is mistake?

6. Why choose cold working 40%?

7. The conclusions of the study should be made clearer and shorter.

Author Response

Thank you for your remarks. Construction criticism is always beneficial to improve the quality of the paper. We attached our statements right beneath your comments, marked as bold text.

  1. There are very few modern studies in the reference. Many of the works cited by the authors were completed more than 20 years ago. It is necessary to analyze modern publications, and there are quite a lot of them. Here are just a few of them:

Another 7 citations from more current papers were added.

  1. Photo in Fig.1 low resolution. It is necessary to improve the quality of the photo and indicate on it the shear bands, grain boundaries.

New samples were prepared and etched in order to improve quality. Figures were replaced.

  1. I recommend combining the diagrams in Fig. 2 (a) and 3 (a) into one Fig. 2 (a) and (b) and bringing them side by side so that the reader can compare them and better see the differences. Combine photos of microstructures in Fig.3.

Remark was accepted and implemented.

  1. In Fig. 4, you need to add indicators of triple points, boundaries of grains of secondary segregations.

Remark was accepted. Description of triple point, grain boundaries and shear bands was added to figure 4 and 6.

  1. The photos in Fig. 4 and 6 are the same, but the captions of the figures are different. This is mistake?

Photos in figure 6 were replaced, thank you for excellent remark. It was a mistake.

  1. Why choose cold working 40%?

In previous research different deformations degrees were evaluated (10%, 20%, 40%). Biggest influence on corrosion resistance of AISI 316L had 40% cold work. This paper is therefore focused on precipitation kinetics of secondary phases at 40% cold work. This research was also realised not only at 650°C but also at 750°C and 900°C. Providing all results would extend the paper too much.

  1. The conclusions of the study should be made clearer and shorter.

Conclusions were reorganised in order to make them short and clear.

Round 2

Reviewer 2 Report

The manuscript has been revised, it can be considered for publication.

Reviewer 3 Report

The authors corrected the paper submitted for review. My comments have been taken into account. The errors in the article have been corrected. I recommend publishing an article in this form.